# *Lactobacillus helveticus R0052* and *Bifidobacterium longum R0175* Supplementation: An Exploratory, Randomized, Placebo-Controlled Trial of Endocannabinoid and Inflammatory Responses in Female Dancers

**DOI:** 10.3390/microorganisms13061284

**Published:** 2025-05-30

**Authors:** Jakub Wiącek, Karolina Skonieczna-Żydecka, Igor Łoniewski, Chariklia K. Deli, Ioannis G. Fatouros, Athanasios Z. Jamurtas, Dominika Moszczyńska, Joanna Karolkiewicz

**Affiliations:** 1Department of Food and Nutrition, Poznan University of Physical Education, 61-871 Poznan, Poland; wiacek@awf.poznan.pl (J.W.); moszczynska@awf.poznan.pl (D.M.); 2Department of Biochemical Research, Pomeranian Medical University, 70-204 Szczecin, Poland; karolina.skonieczna.zydecka@pum.edu.pl; 3Sanprobi Sp. z o.o. Sp.K., 70-535 Szczecin, Poland; sanprobi@sanprobi.pl; 4Department of Physical Education and Sport Science, University of Thessaly, 38221 Trikala, Greece; delixar@pe.uth.gr (C.K.D.); ifatouros@pe.uth.gr (I.G.F.); ajamurt@uth.gr (A.Z.J.)

**Keywords:** endocannabinoid system, gut microbiome, anandamide, lipopolysaccharide, probiotics, dancers

## Abstract

The anandamide (AEA) and lipopolysaccharide (LPS) interaction is gaining attention, but evidence on the influence of probiotics on endocannabinoid system (ECS) biomarkers remains limited. This study (NCT05567653) investigated the effects of 12-week supplementation with *Lactobacillus helveticus* R0052 and *Bifidobacterium longum* R0175 on AEA (main outcome) and inflammatory biomarkers in female dancers. Fifteen participants (5 probiotic, 10 placebo) were included in the final analysis. Serum levels of AEA, LPS, and cytokines (tumor necrosis factor-alpha—TNF-α, interleukin-1 beta—IL-1β, and interleukin-10—IL-10) were measured using an ELISA (enzyme-linked immunosorbent assay), and the psychological stress responses were evaluated using the Mini-COPE questionnaire. At the baseline, a correlation between AEA and LPS was observed (Spearman’s r = 0.9677, *p* < 0.05). After 12 weeks, no statistically significant differences in the AEA, LPS, cytokine levels, or stress-coping strategies were observed between the probiotic and placebo groups (LPS–probiotic: +3.48 EU/L, *p* = 0.9361; placebo: +56.98 EU/L, *p* = 0.0694; AEA–probiotic: −1.11 ng/mL, *p* = 0.9538; placebo: +14.08 ng/mL, *p* = 0.4749). The direction of change may indicate a trend toward increased inflammation in the absence of probiotics, consistent with patterns described in previous literature. However, these results should be viewed as hypothesis generating and warrant confirmation in larger trials.

## 1. Introduction

The endocannabinoid system (ECS) functions via bioactive lipids, mainly anandamide (AEA) and 2-arachidonoylglycerol (2-AG), which activate cannabinoid receptors type 1 (CB1) and type 2 (CB2). CB1 receptors, located primarily in the nervous system, influence pain, mood, sleep, and appetite, whereas CB2 receptors, found mainly on immune cells, are involved in immune regulation [1]. Unlike classical neurotransmitters, endocannabinoids are synthesized on demand in postsynaptic neurons and modulate presynaptic activity by binding to cannabinoid receptors, thus influencing neurotransmitter release [2]. They are not stored in vesicles; instead, their production and degradation rely on tightly regulated enzymatic processes and membrane transporters [3]. Anandamide generally maintains tonic functions, such as emotional stability and appetite regulation under normal conditions, while 2-AG acts in a phasic manner, with levels increasing in response to acute stressors such as injury or public speaking. [4]. Physical activity has been shown to significantly affect ECS activity, particularly by elevating AEA levels [5], which may contribute to the phenomenon known as the ‘runner’s high’, a state of enhanced mood and reduced anxiety following exercise [6].

Physical activity significantly impacts the gut microbiota, microorganisms essential for metabolic regulation, immune function, and neurological processes. Regular exercise can alter microbial composition and abundance [7]. Moderate physical activity enhances gut health by stimulating the production of short-chain fatty acids (SCFAs), particularly butyrate, which supports immune balance and systemic homeostasis [8]. In contrast, excessive exercise without adequate recovery may disrupt the microbial equilibrium, increasing gut permeability and allowing lipopolysaccharide (LPS) translocation into the bloodstream, which can initiate systemic inflammation [9]. LPS, an endotoxin from Gram-negative bacteria, serves as a critical biomarker for microbial translocation and inflammation, reflecting the integrity of the gut barrier. Professional dancers, or artistic athletes, undergo intense training that is linked to a higher injury risk, especially beyond 11.5 h per week. Women are more prone to bone injuries, while men face more contusions and tendinopathies [10,11]. Dancers represent a physiologically and psychologically demanding population, who are frequently exposed to chronic stress. Training sessions are often long and repetitive due to the need to perfect choreography, while performances typically take place in the evening hours and must be reconciled with early morning academic or occupational responsibilities. This irregular schedule contributes to circadian rhythm disruption, which can adversely affect hormonal regulation, immune function, and recovery. Furthermore, performances often involve travel, which increases their exposure to novel pathogens and elevates infection risk. In addition to physical strain, dancers face cognitive demands associated with learning complex motor sequences, as well as performance-related pressure linked to external evaluation by audiences and adjudication panels. Psychological stress has been shown to further increase susceptibility to injury [12], and certain dance forms, such as classical ballet, have been associated with an elevated risk of systemic inflammation [13]. Training despite pain is common in this environment, leading to compensatory movement patterns and reinforcing both biomechanical stress and the psychophysiological burden [14].

The endocannabinoid system (ECS) and gut microbiota interact bidirectionally: microbiota alterations affect ECS-regulated processes, such as inflammation, energy balance, and gut barrier function, while ECS activity can shape microbial composition [15]. This relationship is increasingly studied for its therapeutic potential in gastrointestinal disorders like irritable bowel syndrome (IBS). Rousseaux et al. (2007) found that *Lactobacillus acidophilus* NCFM, at 10^9^ CFU/day, increased cannabinoid receptor expression in intestinal epithelial cells and reduced visceral pain in rats, with effects comparable to morphine [16]. Building on this, Brierley et al. (2023) reviewed the roles of ECS in regard to gastrointestinal function, including motility, secretion, inflammation, and pain perception, and identified it as a target for IBS treatment. However, they noted a lack of robust clinical trials and the limited tolerability of some patients to cannabinoid therapies [17]. Animal studies support a link between anandamide (AEA) and lipopolysaccharide (LPS), suggesting ECS involvement in controlling intestinal permeability and inflammation. Elevated LPS stimulates AEA production in immune cells, highlighting ECS’s regulatory role [18,19,20,21,22,23,24]. Recent studies emphasize the joint influence of ECS, microbiota, and cardiometabolic risk factors [25]. Probiotics, particularly *Lactobacillus* and *Bifidobacterium*, may reduce colonic LPS levels and systemic inflammation, offering a non-pharmacological approach to modulating ECS–microbiota interactions [26,27].

This study used *Lactobacillus helveticus* R0052 and *Bifidobacterium longum* R0175, strains increasingly noted for modulating the nervous system via anti-inflammatory and gut–brain axis pathways [28]. In LPS-challenged rats, they reduced systemic and hippocampal inflammation, improved memory through brain-derived neurotrophic factor (BDNF) regulation, and protected against neurodegeneration by influencing Bax/Bcl-2 expression [29,30]. In stressed Syrian hamsters, these strains shifted gut microbiota and increased IL-10, although anxiety-like behavior remained, possibly due to reduced microbial diversity [31]. In IBS models, their combination reduced visceral hypersensitivity and regulated the hypothalamic–pituitary–adrenal (HPA) axis more effectively than single strains [32]. Early life administration impacted behavior and metabolism in rats on a Western diet, showing sex-specific effects on weight, caloric intake, and leptin levels [33]. In pubertal mice, they mitigated sickness behaviors and LPS-induced immune activation, reducing cytokines in plasma, the prefrontal cortex, and hippocampus in a sex-dependent manner [34]. These strains are among the most studied in regard to mental health and sleep modulation [35,36,37]. With ECS activity closely linked to mood and cognition, current research continues to investigate probiotics with targeted effects on this system. Physical activity, under appropriate recovery conditions, may reduce LPS levels, whereas excessive exertion can have the opposite effect [38,39]. Therefore, our study was conducted during the period of highest physical, mental, and psychological strain for the participants. Although dancers do not train as intensively as elite athletes, such as marathon runners or rugby players, the nature of their workload aligns with the study’s objectives. The study included only female participants, due to a 20-fold higher enrollment rate compared to males. Additionally, probiotic effects may be sex dependent, and including both genders could have introduced variability, complicating data interpretation [40].

Given the limited evidence on cannabinoid-based therapies, particularly among individuals with poor tolerability, alternative approaches to modulating the endocannabinoid system (ECS) warrant investigation. This study examines the effects of probiotic supplementation on anandamide and lipopolysaccharide in physically active individuals. Due to its tonic function and baseline stability, AEA was selected as the primary ECS marker. The secondary outcomes include concentrations of LPS, tumor necrosis factor-alpha (TNF-α), interleukin-1 beta (IL-1β), and interleukin-10 (IL-10), as well as stress-coping strategies, assessed using the Mini-COPE questionnaire. It was hypothesized that probiotic supplementation would enhance gut barrier integrity and attenuate systemic inflammation, resulting in a smaller increase in circulating lipopolysaccharide (LPS) and pro-inflammatory cytokines compared to the placebo. In parallel, the maintenance of gut homeostasis was expected to reduce anandamide (AEA) levels in the probiotic group, in contrast to potential stress-related elevations in the placebo group. In contrast to earlier probiotic studies focused on general inflammation or clinical cohorts, the present trial is the first to directly evaluate probiotic effects on endocannabinoid markers in a physically active population. By examining female dancers, a unique athletic cohort under high training and stress load, our study extends previous research in regard to a new context, linking probiotic supplementation to the endocannabinoid–inflammation axis in a way that has not been explored before.

## 2. Materials and Methods

### 2.1. Participants

The sample size calculations were conducted using G*Power software (version 3.1.9.7; Heinrich Heine University Düsseldorf, Germany). To achieve 80% statistical power to detect a medium effect size (Cohen’s d = 0.5) in the primary outcome variable (anandamide, AEA), with a two-tailed α set at 0.05 and based on the expected variability, approximately 51 participants per group was required. The remaining biomarkers, including LPS, IL-1β, IL-10, and TNF-α, were measured at the same time points (before and after the 12-week intervention), as secondary outcomes. There were no changes in the trial methodology or outcomes throughout the study.

The study included participants aged 18 to 36 years, who were actively engaged in professional dancing, with their training exceeding 8 h per week. Individuals were excluded if, within the three months preceding the study, they had sustained injuries, consumed prebiotics and/or probiotics, been hospitalized, traveled to tropical countries, or used antibiotics, cannabis products, steroids, or anabolic steroids. Individuals with chronic diseases requiring ongoing pharmacological treatment, including metabolic, inflammatory, autoimmune, or psychiatric conditions, were excluded from participation to minimize confounding influences on inflammatory and neuroimmune markers.

The study included female dance students from the Academy of Physical Education, conducted during the busiest period of the semester, with the highest academic workload. Their schedule consisted of theoretical lectures, motor preparation training, recreational movement classes, gym sessions, choreography training, and teaching practice in dance schools. Furthermore, some participants worked professionally in theaters, including weekend performances. The demanding and irregular timetable (sometimes from 8 AM to 8 PM), combined with the stress of learning choreography, public performances, and exam preparation, contributed to the overall physical and psychological burden experienced.

Despite careful participant selection based on anthropometric measurements and activity levels, the target group size was not achieved. In addition to the dropout reasons listed below, several challenges contributed to recruitment difficulties, including a lack of male participants (only two men volunteered), the evening training and performance schedules of the dancers, which often led to late wake-up times, conflicting with morning sample collection hours, logistical difficulties related to fecal sample collection required for other procedures, and the inability to ensure participants would maintain a consistent diet throughout the intervention.

### 2.2. Intervention

Blinding was applied to both the study authors and the participants. Randomization (block randomization with a block size of four) was conducted using a computer-generated algorithm (www.randomizer.org (accessed on 25 September 2022)) by the probiotic manufacturer. The randomization was performed by the manufacturer of the probiotic and placebo used in the study. The probiotic group received *Lactobacillus helveticus* R0052 (CNCM strain I-1722) and *Bifidobacterium longum* R0175 (CNCM strain I-3470) (trade name: Sanprobi Stress; manufacturer: Sanprobi sp. z o. o. sp. k., Szczecin, Poland; lyophilisate: Lallemand Health Solutions Inc., Mirabel, Quebec, Canada) at a dose of 3 × 10^9^ colony forming units (CFUs)/active fluorescent units (AFU) per capsule, taken daily, in the morning, for 12 weeks. The placebo capsules, identical in terms of mass and appearance to the probiotic capsules, contained maltodextrin and corn starch as carriers. Both types of capsules were identical in appearance.

The interventions were well-tolerated, with no adverse effects reported, and the probiotic product adhered to safety certifications. From an initial pool of 51 volunteers, 26 female participants met the inclusion criteria. However, only twenty attended the blood and stool sample collection, and three subsequently failed to collect the study supplement, providing no explanation for their absence. Seventeen female dancers completed the study; however, one participant was excluded during the statistical analysis due to an outlier body mass index (BMI) indicative of overweight status, significantly differing from other group members. This exclusion ensured sample homogeneity and the validity of the results, as BMI is a potential confounding factor in probiotic health studies. Sixteen participants proceeded to the final analysis, with eleven receiving the placebo (PLA) and five receiving the probiotic (PRO) treatment. During the preliminary statistical analysis of the primary outcome, anandamide (AEA) levels, an additional outlier was identified. One data point exhibited a substantial deviation from the overall distribution, resulting in the final group having 15 participants. We conducted our analysis on a per-protocol basis, including only those participants who completed the trial (N = 15). To monitor adherence to the probiotic/placebo regimen, participants were asked to maintain daily logs of their capsule intake and return all the unused capsules at their follow-up visit. Compliance was verified by counting the returned capsules. Adherence to the intervention was 100%, with participants consuming the doses corresponding to the full 12-week period (84 days).

### 2.3. Study Protocol

All the procedures were conducted in the laboratories at Poznan University of Physical Education, in accordance with established research protocols and ethical guidelines. Recruitment began in October 2022 and concluded in March 2023. Further sample collection was not possible due to the limited number of willing participants who met the study criteria. The study would have been discontinued if a significant number of participants had chosen to withdraw from the intervention. During an informational session, the participants were instructed to maintain their habitual dietary patterns throughout the study, while diligently reporting any dietary changes, the emergence of exclusion criteria, or potential adverse effects. The participants were also advised to attend the blood sampling session in a rested state, following a light dinner and a 2–3 day pause from training activities. One week prior to the trial, anthropometric measurements were conducted using bioelectrical impedance analysis (BIA), and dietary data were collected. Fasting blood samples (10 mL) were collected one day prior to the initiation of supplementation with either the probiotic or placebo and again after 3 months (12 weeks). Blood samples were centrifuged to separate the serum. The serum samples were immediately frozen and stored for subsequent analysis using enzyme-linked immunosorbent assays (ELISAs), while morphological analysis was performed directly following collection.

The study was registered as a clinical trial (www.clinicaltrials.gov (accessed on 22 September 2022); NCT05567653—Effects of Probiotics on Gut Microbiota, Endocannabinoid and Immune Activation and Symptoms of Fatigue in Dancers), complied with the Declaration of Helsinki, and received approval from the Bioethics Committee at the Poznan University of Medical Sciences (approval no. 412/22). All the study participants provided their written informed consent. The report follows the CONSORT 2010 guidelines, with a flowchart of the participant inclusion process presented in Figure 1.

### 2.4. Methods

#### 2.4.1. Outcome Measures

To investigate the interaction between the endocannabinoid system and immune function, serum concentrations of anandamide (AEA) and lipopolysaccharides (LPS) were quantified using enzyme-linked immunosorbent assays (double-antibody sandwich ELISA; SunRed Biotechnology Co., Ltd., Shanghai, China). The assay sensitivity and detection range were as follows: AEA (7.125 ng/mL; 8–2000 ng/mL) and LPS (10.725 EU/L; 12–4000 EU/L). The inflammatory status was assessed by measuring the serum concentrations of TNF-α, IL-1β, and IL-10 using an ELISA (SunRed Biotechnology Company), with assay sensitivity and detection ranges of TNF-α (2.827 ng/mL; 3–900 ng/mL), IL-1β (15.013 pg/mL; 20–8000 pg/mL), and IL-10 (9.012 pg/mL; 10–3000 pg/mL). According to the manufacturer, the intra-assay coefficient of variation (CV) is less than 10%, and the inter-assay CV is below 12%. Therefore, minor fluctuations in biomarker concentrations may lie within the assay’s variability range, potentially masking subtle effects of the intervention.

The Mini-COPE questionnaire was used to assess dispositional stress-coping mecha-nisms, evaluating typical emotional and behavioral responses under high-stress conditions. It consists of 28 items, adapted from the original 60-item COPE inventory, designed to measure habitual ways of reacting to difficult situations. The participants responded using a four-point Likert scale, ranging from “almost never” to “almost always.” The tool distinguishes 14 coping strategies, grouped into three major factors: Active Coping, Avoidance Behaviors, and Seeking Support/Emotion-Focused Coping, with maximum scores of 18, 30, and 36 points, respectively. Body composition was assessed using bioelectrical impedance analysis with a Seca analyzer (Seca GmbH & Co., Hamburg, Germany), providing detailed metrics on body mass, fat percentage, and lean body mass. Nutritional status was evaluated through the analysis of 3-day dietary records, assessing energy intake, macronutrient distribution, and dietary patterns. Nutritional status was also evaluated using 3-day dietary records and the 14-Item Mediterranean Diet Assessment Tool (MDAT), which measures adherence to a diet high in fruits, vegetables, legumes, whole grains, fish, and olive oil, while restricting refined grains, sugar-sweetened beverages, and trans fats. These factors are particularly relevant as a person’s BMI may influence components of the endocannabinoid system (ECS); cholesterol-rich foods are a source of arachidonic acid, a precursor for anandamide (AEA), and dietary fiber modulates the gut microbiome. Blood morphology, including white blood cell (WBC) and lymphocyte counts, was analyzed using flow cytometry with the Synergy 2 SIAFRT analyzer (BioTek, Winooski, VT, USA), providing insights into immune function and systemic health. These data on body composition, dietary habits, and blood morphology were used to ensure the homogeneity of the study group.

Statistical calculations and primary outcome data, including group characteristics, dietary intake, biomarker measurements, Mediterranean diet adherence scores, and Mini-COPE responses, are provided in the Appendix A.

#### 2.4.2. Statistical Analysis

Statistical analysis was performed using the STATISTICA 13.3 (TIBCO Software Inc., Palo Alto, CA, USA) package. For comparing the baseline data between the groups, the *t*-test or Mann–Whitney U test was used if the normality condition was not met. The Shapiro–Wilk test was used to check the data for the normality of the distribution. To evaluate the correlation between the levels of AEA and LPS, Spearman’s rank-order correlation was employed. To compare variables after 3 months of supplementation with either the probiotic or the placebo, the *t*-test or Wilcoxon test was used if the normality condition was not met. A two-way analysis of variance (ANOVA) for repeated measures was used to analyze the differences in regard to the effect of time, group and time x group. For all the statistical tests, a *p* value of less than 0.05 was interpreted as statistically significant. For statistically significant results, the coefficients η2 are presented as an indicator of the effect size. If statistically significant results had been observed, post hoc analyses with Bonferroni correction would have been used to adjust for multiple comparisons, ensuring robustness against Type I errors. Moreover, effect size measures (e.g., Cohen’s d for parametric tests or rank-biserial correlation for non-parametric tests) would have been calculated to better interpret the magnitude of the observed effects.

## 3. Results

### 3.1. No Baseline Differences in Anthropometric or Dietary Variables Between the Groups

The baseline measurements (BMI, fat percentage, white blood cells, and lymphocyte counts), used to standardize the study groups, are presented in Table 1. The dancers were recruited from two groups with similar skill levels, following an identical training regimen. The comparative analysis of the participants’ diets revealed no significant differences between the groups. The characteristics of the diet are also presented in Table 1.

The baseline data collected indicate favorable and methodologically sound conditions for conducting comparative analyses between the study groups and minimizing the potential confounding effects in subsequent analyses.

### 3.2. No Statistically Significant Changes in AEA, LPS, or Cytokine Levels Following Supplementation

#### 3.2.1. Strong Positive Correlation Between AEA and LPS at the Baseline

In accordance with the expectations derived from the literature review available on the discussion, the levels of AEA and LPS were found to be correlated (r = 0.9677; *p* < 0.05), as assessed by Spearman’s rank-order correlation. This correlation is illustrated in Figure 2. The clearly visible outlier on the graph was excluded from further analyses.

#### 3.2.2. Directional Trends Observed, but Between- and Within-Group Changes Were Not Significant

After homogenizing the groups by excluding the outliers, the normality of the data distribution was assessed using the Shapiro–Wilk test, confirming that there were no significant deviations from normality (*p* > 0.05) for all the variables (*n* = 5 for the probiotic group; *n* = 10 for the placebo group). At the baseline, the probiotic and placebo groups did not differ significantly in regard to any of the measured biomarkers (LPS (505.05 vs. 579.36 EU/L, *p* = 0.3391), AEA (253.99 vs. 292.59 ng/mL, *p* = 0.3329)). A comparison of the baseline levels of the analyzed indicators between the groups is presented in Table 2.

At the endpoint, the delta values did not indicate a significant magnitude in the difference between the main biomarkers of interest: LPS (probiotic: 3.48, *p* = 0.9361; placebo: 56.98, *p* = 0.0694), AEA (probiotic: −1.11, *p* = 0.9538; placebo: 14.08, *p* = 0.4749), and other measured biomarkers. The data distribution within the groups remained consistent, except for IL-10 in the probiotic group, wherein a shift was observed (Table 3).

To visualize individual-level variations and group-wise trends, the key outcome data originally presented in Table 2 and Table 3 are shown as column scatter plots (Figure 3).

Notably, AEA in the probiotic group was the only parameter with a negative delta. A comparison of the post-intervention results and the differences relative to the baseline for both groups are presented in Table 4.

### 3.3. No Effect of Probiotic Supplementation on Stress-Coping Strategies

Similar to anandamide and the inflammatory markers, no statistically significant changes were observed in the stress-coping strategies assessed using the Mini-COPE questionnaire, and the direction and magnitude of such changes did not suggest any clinically meaningful effect. The results of this analysis are presented in Table 5.

## 4. Discussion

Although a greater mean increase in lipopolysaccharide (LPS) levels was observed in the placebo group compared to the probiotic group, and the mean anandamide (AEA) levels decreased in the probiotic group, while slightly increasing in the placebo group, these patterns did not reach statistical significance. Therefore, the hypothesis proposed in the Introduction was not supported by the results. To date, no studies have investigated the effects of probiotic therapy on endocannabinoid system markers, particularly in physically active populations. One randomized, double-blind trial in 92 coronary artery disease patients examined symbiotic supplementation combining inulin and *Lactobacillus rhamnosus* GG, observing reductions in lipopolysaccharides (LPS) and inflammatory markers, as well as a correlation between CB2 receptor expression and LPS levels, indicating improved gut barrier function [41]. However, the study population and intervention differed substantially to ours. One of the methodological limitations of our study was the use of an ELISA to quantify the AEA levels, which is less specific and less sensitive than liquid chromatography–mass spectrometry (LC–MS). Future studies should consider employing LC–MS for a more accurate assessment of endocannabinoid concentrations.

A number of meta-analyses have demonstrated significant anti-inflammatory effects of probiotics in clinical and general populations. For instance, a meta-analysis of 26 randomized controlled trials reported a significant reduction in LPS (SMD –0.47, 95% CI –0.85 to –0.09, *p* = 0.02), along with improvements in gut barrier function and inflammatory markers, such as the C-reactive protein (CRP), TNF-α, and IL-6 [42]. Another review of 11 trials confirmed reductions in TNF-α, IL-6, IL-12, and IL-4, and increases in IL-10 [43]. Importantly, a meta-analysis focused on elite athletes found that probiotic supplementation significantly reduced TNF-α concentrations (–2.31 pg/mL; 95% CI –4.12, –0.51 pg/mL; *p* = 0.01), suggesting potential relevance in physically active individuals [44]. However, the strain specificity and heterogeneity across studies make it difficult to generalize these results to the present trial. In our study, no significant changes in the stress-coping strategies (Mini-COPE) were observed in either group over the study period, suggesting that probiotic supplementation did not meaningfully influence coping mechanisms. Chronic stress disrupts gut microbiota, leading to dysbiosis and intestinal barrier damage, creating a cycle where gut dysfunction amplifies stress responses [45]. An increase in blood LPS levels (“endotoxemia”) activates the immune system, triggering the release of pro-inflammatory cytokines (TNF-α, IL-1β, IL-6), which can cross the blood–brain barrier, leading to neuroinflammation, HPA axis activation, and behavioral symptoms, such as anhedonia, anxiety, and low mood, resembling depression or post-traumatic stress disorder (PTSD), as demonstrated in animal studies wherein LPS induced depression- and anxiety-like behaviors [46]. However, in our study, stress levels were likely lower than in elite endurance athletes or individuals with diagnosed depression, anxiety, or chronic diseases, potentially affecting the magnitude of these effects. A potential explanation is the lack of a significant reduction in lipopolysaccharide (LPS) levels in the probiotic group. Research on *Lactobacillus helveticus* R0052 and *Bifidobacterium longum* R0175 remains mixed. One randomized trial combining animal and human data showed that this probiotic formulation reduced anxiety-like behavior in rats and improved psychological distress and coping strategies in humans [47]. In contrast, another study in individuals with low mood reported no significant changes in mood or inflammatory markers [48], while a separate trial in patients with major depressive disorder found reductions in depression scores, alongside changes in tryptophan metabolism, suggesting potential neuroimmune modulation [49]. In vitro studies using the SHIME^®^ model also support the anti-inflammatory and gut-modulatory potential of these strains [50]. The systematic review by Heimer et al. (2022) analyzed the effects of probiotics on immune function, upper respiratory tract infections (URTI), and gastrointestinal (GI) symptoms in athletes and active individuals. The findings were inconclusive, highlighting inconsistencies across studies regarding probiotic strains, delivery methods, participant performance levels, treatment durations, and outcome assessments [51]. These findings are relevant to our study; the observed variability in probiotic efficacy across different studies and populations may explain the lack of significant changes in our measured outcomes.

### Limitations

Given that all the participants were female, the findings of this study cannot be generalized to male populations. Sex-dependent differences in the probiotic response and ECS function suggest that future studies should include both sexes to explore potential variability. Despite efforts to recruit from multiple dance schools, performance groups, and theaters, logistical barriers and insufficient male enrollment prevented us from reaching this target. According to our pre-study power analysis, more than 100 participants would have been needed to reliably detect such effects. Therefore, all the findings should be interpreted as exploratory and preliminary.

No follow-up measurements were performed, limiting the ability to evaluate the sustainability of the observed changes over time. The study design also lacked groups receiving varying doses of the probiotics or supplementation combined with prebiotics, which could have provided insights into dose–response relationships and synergistic effects. Two randomized participants were excluded from the analysis (one due to a BMI outlier and one due to an extreme AEA value identified as an outlier). While these exclusions were made to maintain group homogeneity and data quality, we acknowledge that removing participants post-randomization can introduce bias. The lack of an intention-to-treat analysis is a limitation of our study design. The study did not include a sedentary or low-activity control group, which limits the ability to distinguish the effects of the probiotic intervention from those related to a high level of physical activity.

It is worth noting that we did not analyze the gut microbiota composition in this trial. Therefore, any discussion about the microbiota-mediated mechanisms underlying our results remains speculative. Additionally, dietary intake was assessed only at the baseline. No method was available to monitor specific dietary components, such as arachidonic acid intake, which may influence endogenous anandamide (AEA) synthesis and could act as a potential confounding factor. However, the participants were required to agree not to introduce major dietary changes during the intervention and to report any deviations, but no continuous dietary monitoring was performed throughout the study period.

Anandamide (AEA) concentrations were measured using a commercial ELISA kit. We acknowledge that ELISA, while convenient, is not the gold standard for endocannabinoid quantification. The sensitivity and specificity of ELISA are lower than those of liquid chromatography–mass spectrometry (LC–MS/MS), which is the preferred method for precise AEA measurement. The selection of the ELISA was driven by financial constraints and the technical capacity of the research facility at the time. The ELISA-based methods used to quantify AEA, LPS, and cytokines are subject to inherent analytical variability; according to the manufacturer, the intra-assay CV was <10% and the inter-assay CV was <12%, suggesting that small fluctuations in the measured concentrations may fall within the scope of assay error and should be interpreted with caution.

Although one of the co-authors holds shares in the company producing the probiotic used in this study, and another receives financial compensation from the same entity, this relationship did not influence the design, statistical analysis, interpretation, or reporting of the findings. Blinding and data management procedures were designed to maintain objectivity and independence.

## 5. Conclusions

In our study, no significant impact of *Lactobacillus helveticus* R0052 and *Bifidobacterium longum* R0175 supplementation was observed on the levels of anandamide (AEA), lipopolysaccharide (LPS), or inflammatory markers in professional dancers. Although the study followed a rigorous randomized placebo-controlled design, the small sample size limits the generalizability and interpretability of the findings. Our findings should be viewed as exploratory and hypothesis generating. While prior research supports the anti-inflammatory potential of these strains, the results indicate that their efficacy may be context dependent in physically active populations. Further studies, with larger sample sizes and varied intervention designs, are warranted to explore these interactions more comprehensively.

## Figures and Tables

**Figure 1 microorganisms-13-01284-f001:**
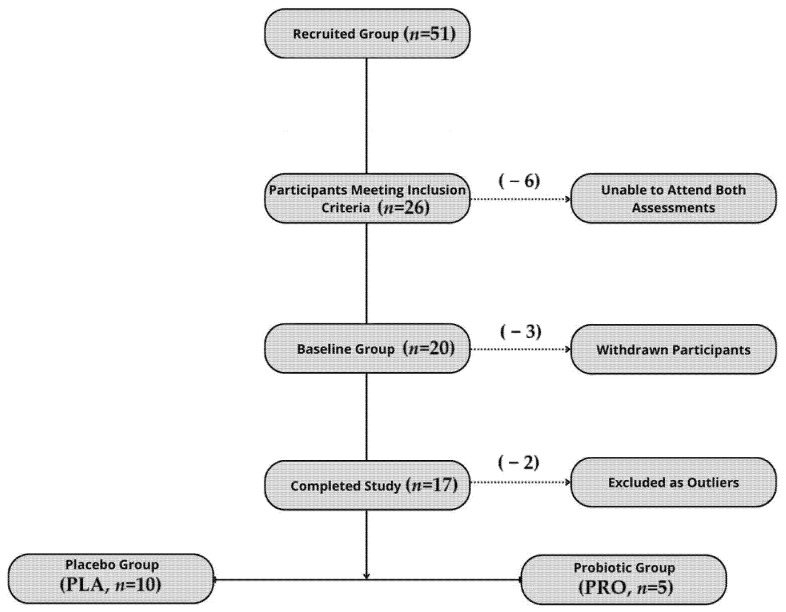
Flowchart illustrating participant recruitment, inclusion, dropout, and final allocation to study groups.

**Figure 2 microorganisms-13-01284-f002:**
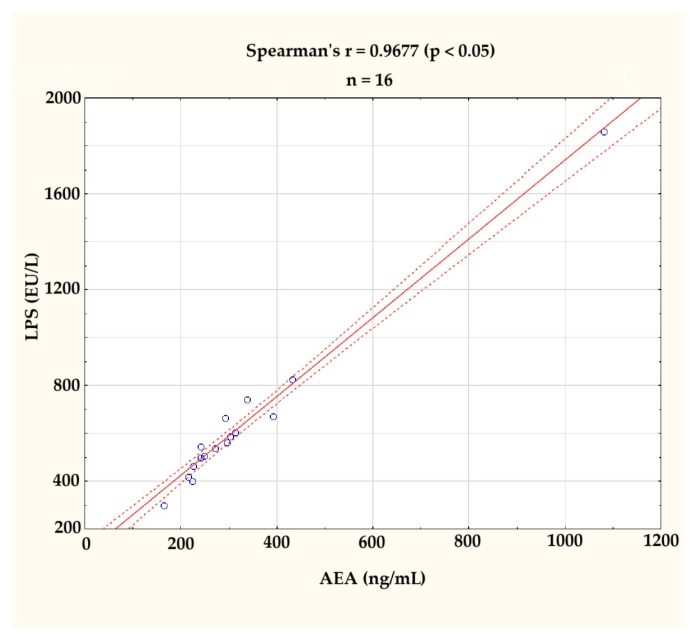
Correlation between serum anandamide (AEA, ng/mL) and lipopolysaccharide (LPS, EU/L) levels at the baseline (*n* = 16). A strong positive correlation was observed (Spearman’s r = 0.9677, *p* < 0.05). The figure displays individual values (blue circles), a fitted regression line (solid red line), and a 95% confidence interval (dashed red lines). One outlier, identified visually and statistically, was excluded from further analysis.

**Figure 3 microorganisms-13-01284-f003:**
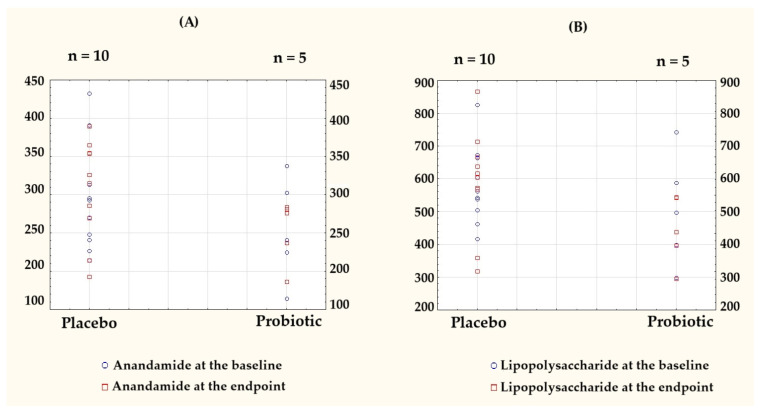
Individual-level data presented as column scatter plots to illustrate the baseline and endpoint values for (**A**) anandamide (AEA) and (**B**) lipopolysaccharide (LPS) in the placebo (*n* = 10) and probiotic (*n* = 5) groups. Each dot represents a single participant, plotted separately for each group to visualize the within-group distributions and changes over time. Baseline values are marked with blue circles, endpoint values with red squares.

**Table 1 microorganisms-13-01284-t001:** Baseline characteristics (anthropometrics and diet). All participants were female.

	Probiotic (*n* = 5)	Placebo (*n* = 11)	Total (*n* = 16)	Placebo vs. Probiotic (*p* Value)
	Mean ± SD (Min–Max)	Mean ± SD (Min–Max)	Mean ± SD (Min–Max)	
Dancing activity [hours per week]	16.00 ± 9.77 (9.50–29.00)	17.11 ± 6.98 (8.00–33.00)	16.77 ± 7.63 (8.00–33.00)	0.69 ^b^
Age [years]	20.00 ± 1.30 (19–22)	20.55 ± 1.04 (19–22)	20.44 ± 1.09 (19–22)	0.55 ^b^
Body mass [kg]	60.10 ± 7.31 (48.60–68.30)	58.07 ± 6.95 (49.40–68.70)	58.08 ± 6.81 (48.60–68.70)	0.99 ^a^
BMI (body mass index) [kg/m^2^]	20.80 ± 2.29 (18.10–25.10)	21.05 ± 2.18 (17.70–23.40)	21.02 ± 2.13 (17.70–25.10)	0.93 ^a^
Fat [% body mass]	27 ± 3 (25–31)	27 ± 4 (21–31)	27 ± 3 (21–31)	0.84 ^a^
WBC [×10^9^/L]	5.94 ± 1.17 (4.5–6.6)	5.54 ± 0.84 (4.0–7.5)	5.66 ± 1.07 (4.0–7.5)	0.50 ^a^
Lymphocytes [×10^9^/L]	2.60 ± 0.46 (1.6–3.3)	2.34 ± 0.79 (1.8–3.4)	2.42 ± 0.57 (1.6–3.4)	0.41 ^a^
Energy [kcal]	2325.54 ± 425.00 (1835.0–2842.6)	1999.23 ± 279.81 (1588.93–2578.45)	2101.20 ± 353.22 (1588.93–2842.6)	0.26 ^a^
Protein [g]	100.47 ± 21.09 (79.42–130.92)	85.29 ± 30.13 (49.52–154.26)	90.03 ± 27.87 (49.52–154.26)	0.51 ^a^
Fat [g]	90.47 ± 18.72 (67.02–115.73)	74.56 ± 13.93 (57.69–95.74)	79.54 ± 16.76 (57.69–115.73)	0.41 ^a^
Cholesterol [mg]	330.21 ± 140.77 (178.32–490.73)	216.73 ± 104.36 (9–372.03)	252.19 ± 124.49 (9–490.73)	0.40 ^a^
Carbohydrates [g]	298.79 ± 57.98 (240.54–376.75)	271.12 ± 53.14 (194.89–359.42)	279.77 ± 54.36 (194.89–376.75)	0.75 ^a^
Fiber [g]	21.36 ± 12.67 (15.03–27.63)	28.96 ± 15.13 (16.69–47.68)	26.59 ± 14.44 (15.03–47.68)	0.91 ^b^
Mediterranean diet adherence [0–14]	5.46 ± 1.86 (3.00–10.00)	6.20 ± 2.49 (3.00–10.00)	5.69 ± 2.02 (3.00–10.00)	0.42 ^a^

Note: ^a^—*t* test/^b^—Mann–Whitney U test.

**Table 2 microorganisms-13-01284-t002:** Between-group analysis at the baseline.

Mean	Probiotic Group (*n* = 5)		Placebo Group (*n* = 10)		(Independent *t*-Test *p*)
	Baseline (Mean ± SD) (Min–Max)	(Shapiro–Wilk *p*)	Baseline (Mean ± SD) (Min–Max)	(Shapiro–Wilk *p*)	
LPS [EU/L*]	505.05 ± 170.77 299.35–742.17	0.9796	579.36 ± 118.50 416.63–826.21	0.6987	0.3391
AEA [ng/mL]	253.99 ± 68,01 164.05–337.67	0.8996	292.59 ± 70.98 214.40–432.66	0.1966	0.3329
TNF-α [pg/mL]	87.52 ± 24.40 56.75–117.74	0.9029	105.89 ± 22.23 72.16–139.53	0.8120	0.1671
IL-1β [pg/mL]	1300.42 ± 298.48 982.44−1689.51	0.4774	1475.79 ± 333.89 940.79–2090.87	0.9272	0.3402
IL-10 [pg/mL]	490.03 ± 110.89 393.35–679.28	0.1302	515.46 ± 96.86 397.89–709.14	0.6118	0.6546

* Endotoxin units per liter.

**Table 3 microorganisms-13-01284-t003:** Between-group analysis at the endpoint.

Mean	Probiotic Group (*n* = 5)		Placebo Group (*n* = 10)		(Independent *t*-Test *p*)
	Endpoint (Mean ± SD) (Min–Max)	(Shapiro–Wilk *p*)	Endpoint (Mean ± SD) (Min–Max)	(Shapiro–Wilk *p*)	
LPS [EU/L]	508.53 ± 89.83 382.19–594.96	0.4817	636.34 ± 136.85 401.08–871.60	0.4620	0.1210
AEA [ng/mL]	252.83 ± 41.75 186.22–283.97	0.1123	306.67 ± 65.45 192.98–389.58	0.4332	0.0831
TNF-α [pg/mL]	97.81 ± 17.51 68.48–115.42	0.1965	113.04 ± 22.80 76.75–146.56	0.7815	0.2144
IL-1β [pg/mL]	1346.94 ± 209.28 1001.15–1509.79	0.1242	1580.40 ± 325.51 1077.65–2183.22	0.8382	0.1717
IL-10 [pg/mL]	556.20 ± 31.95 500.76–583.24	0.0387	540.40 ± 81.99 395.87–663.44	0.9436	1.0000 *

* Mann–Whitney U test.

**Table 4 microorganisms-13-01284-t004:** Within-group analysis at the endpoint.

Mean	Probiotic Group (*n* = 5)		Placebo Group (*n* = 10)	
	Delta	Paired *t*-Test (t, *p*)	Delta	Paired *t*-Test (t; *p*)
LPS [EU/L]	3.48	0.0853; 0.9361	56.98	2.0605; 0.0694
AEA [ng/mL]	−1.11	−0.0616; 0.9538	14.08	0.7457; 0.4749
TNF-α [pg/mL]	10.28	1.8887; 0.1320	7.15	1.2768; 0.2336
IL-1β [pg/mL]	46.52	0.5646; 0.6025	104.61	1.9067; 0.0889
IL-10 [pg/mL]	66.17	0.9439; 0.3452 *	24.95	1.1912; 0.2640

* Wilcoxon signed-rank test (Z; *p*).

**Table 5 microorganisms-13-01284-t005:** Effects of probiotic supplementation on stress-coping strategies assessed using the Mini-COPE questionnaire before supplementation and after 3 months.

	Probiotic Group (*n* = 5) Mean ± SD				Placebo Group (*n* = 10) Mean ± SD			
	Pre (Mean ± SD) (Min–Max)	Shapiro–Wilk *p*	Post (Mean ± SD) (Min–Max)	Paired *t*-Test (t; *p*)	Pre (Mean ± SD) (Min–Max)	Shapiro–Wilk *p*	Post (Mean ± SD) (Min–Max)	Paired *t*-Test (t; *p*)
Active Coping [0–18]	13 ± 4.53 8–18	0.3946	12 ± 3.94 9–18	0.9535; 0.3943	12.8 ± 3.71 6–18	0.5996	12 ± 3.97 4–17	0.7530; 0.4707
Avoidance Behaviors [0–30]	14.6 ± 2.61 13–19	0.0214	13 ± 5 6–20	0.3652; 0.7150 *	14.8 ± 4.73 8–22	0.7620	13.2 ± 5.07 4–20	0.9130; 0.3850
Seeking Support/Emotion-Focused Coping [0–36]	16.4 ± 1.67 14–18	0.3140	16 ± 2.92 14–21	0.4313; 0.6885	16.7 ± 4.45 11–24	0.3767	15.5 ± 4.06 9–23	1.1078; 0.2967

* Wilcoxon signed-rank test (Z; *p*).

## Data Availability

The data presented in this study are openly available in Zenodo at [https://doi.org/10.5281/zenodo.15039219]. Additionally, data are also included in the Appendix A.

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
