# Peer review of "Lactobacillus helveticus R0052 and Bifidobacterium longum R0175 Supplementation: An Exploratory, Randomized, Placebo-Controlled Trial of Endocannabinoid and Inflammatory Responses in Female Dancers"

_microorganisms, 2025, doi:10.3390/microorganisms13061284_

Round 1
Reviewer 1 Report
Comments and Suggestions for Authors
The article titled "Assessment of Endocannabinoid and Inflammatory Markers in Dancers Following 12-Week Supplementation with Lactobacillus helveticus R0052 and Bifidobacterium longum R0175 – a randomized, placebo-controlled trial" investigates the effects of the probiotics Lactobacillus helveticus R0052 and Bifidobacterium longum R0175 on the endocannabinoid system marker anandamide (AEA), the LPS system, cytokine production involved in inflammation (such as TNF-α, IL-1β, IL-10), and psychological coping strategies in professional dancers. The study was randomized and placebo-controlled, with 15 participants in the final analysis (5 in the probiotic group and 10 in the placebo group). The results show that although the probiotic group exhibited a slight reduction in AEA and a minimal increase in LPS (opposite to the placebo group), none of the changes in biomarkers or psychological questionnaires reached statistical significance. The authors conclude that the effects of probiotics on the endocannabinoid system and inflammation in physically active populations may be highly context-dependent and influenced by sample size.
Overall, the manuscript is clear and relevant to the field, and it is presented in a well-structured manner. The cited publications are recent and consistent with the scope of the study. The proposed research is generally valid, and the experimental design is appropriate. Nevertheless, the manuscript has some shortcomings, which are outlined in the points below:
- The overall text is well written; however, considering the title, the authors are advised to make it more concise.
- The final sample size (n=15) is too limited to draw reliable conclusions. The authors are encouraged to strengthen the Limitations section by emphasizing that the statistical power was well below what is needed to detect medium-sized effects (Cohen’s d = 0.5 would require >100 participants). The small sample size seriously compromises the statistical power and the validity of the conclusions.
- The authors measured AEA levels using ELISA assy, which is less precise than liquid chromatography–mass spectrometry (LC-MS). It is recommended that the authors acknowledge this methodological limitation in the Discussion section and suggest LC-MS as a preferred technique for future studies.
- All participants were female. Although this is justified by low male enrollment, the authors should explicitly state that the results cannot be generalized to male populations.
- One of the authors is a shareholder in the company producing the probiotic. While this is disclosed, it is somewhat downplayed. It is recommended to also address this in the Discussion section, reaffirming that it did not influence data analysis or interpretation.
The tables are appropriate, easy to understand, and the data are consistent with the manuscript; however, minor corrections are needed:
- In the caption of Table 2, adjust the font style (the phrase “Table 2” on line 293 appears in a different font compared to the other table captions).
- Check the font style in Table 5 as well
- change "regula-tion" (line 89) to "regulation" and "supple-mentation" (line 113) to "supplementation".
The figures lack detailed captions. The authors should enhance the visual quality of the graphs (e.g., by improving axis label readability) and include more informative figure legends.
The study is interesting and methodologically sound in terms of randomization and placebo control. However, it is heavily limited by the small sample size. The conclusions should be presented with more caution, and the observed trends deserve further investigation in larger-scale studies using more sensitive measurement tools.
Author Response
We thank the Reviewer for their thorough evaluation of our manuscript and for providing constructive and insightful feedback. We have addressed each point carefully and detail our responses below.
Comment 1: The overall text is well written; however, considering the title, the authors are advised to make it more concise.
- The title has been shortened to improve clarity and better reflect the study's focus without unnecessary length (page 1, line 2).
Comment 2: The final sample size (n=15) is too limited to draw reliable conclusions. The authors are encouraged to strengthen the Limitations section by emphasizing that the statistical power was well below what is needed to detect medium-sized effects (Cohen’s d = 0.5 would require >100 participants).
- We have revised the Limitations section to explicitly state that the final sample size severely limited statistical power. We now include a reference to power calculations indicating that more than 100 participants would have been needed to detect medium effects (page 15, line 519).
Comment 3: The authors measured AEA levels using ELISA assay, which is less precise than LC-MS. It is recommended that the authors acknowledge this methodological limitation in the Discussion section and suggest LC-MS as a preferred technique for future studies.
- We agree with the reviewer that LC-MS would offer more accurate quantification of endocannabinoid concentrations. The Discussion has been updated to recognize that ELISA lacks the sensitivity and specificity of LC-MS, and we now recommend the latter for future studies (page 14, line 441).
Comment 4: All participants were female. Although this is justified by low male enrollment, the authors should explicitly state that the results cannot be generalized to male populations.
- We have clarified this in the Discussion. While the all-female sample was a result of recruitment feasibility, we now explicitly state that the findings may not apply to males and that sex-specific responses should be explored in future studies (page 15, line 516).
Comment 5: One of the authors is a shareholder in the company producing the probiotic. While this is disclosed, it is somewhat downplayed. It is recommended to also address this in the Discussion section, reaffirming that it did not influence data analysis or interpretation.
- We have added a clarification in the Discussion to emphasize that this affiliation had no influence on the study's design, conduct, or interpretation of results. This supplements the formal conflict of interest disclosure and ensures full transparency (page 16, line 554).
Comment 6: Minor corrections are needed in tables and typography:
- These typographical and formatting inconsistencies have been corrected. Font styles in Table captions have been unified, and all improper hyphenation has been eliminated in the revised manuscript.
Comment 7: The figures lack detailed captions. The authors should enhance the visual quality of the graphs (e.g., by improving axis label readability) and include more informative figure legends.
- Figure 2 has been substantially revised. The updated version now includes clearly labeled axes with appropriate units (AEA [ng/mL] and LPS [EU/L]), and a refined layout. The figure now displays a scatter plot illustrating the correlation between serum anandamide and lipopolysaccharide levels, with a linear regression line and 95% confidence interval band. The statistical parameters (Spearman’s r and p-value) are presented directly above the plot, enhancing clarity and interpretability. Furthermore, the figure legend has been rewritten to describe the content and relevance of the graph in greater detail.
Comment 8: The study is interesting and methodologically sound in terms of randomization and placebo control. However, it is heavily limited by the small sample size. The conclusions should be presented with more caution, and the observed trends deserve further investigation in larger-scale studies using more sensitive measurement tools.
- The Discussion and Conclusion sections have been updated to reflect the exploratory nature of the results. We now describe the findings as preliminary and highlight the need for larger, better-powered studies that incorporate more sensitive assays and broader samples.
Reviewer 2 Report
Comments and Suggestions for Authors
- The authors should describe how their work is unique and how it differs from previous studies.
- The statement "The sample size in this study was insufficient to achieve the statistical power required for detecting meaningful effects" appears on page 11 of line # 405. Why wasn't this issue taken into account before the current study was conducted?
- On page 7 of line # 278, please replace Figure 1 with Figure 2.
- Please include the X and Y axis labels and values in Figure 2.
- Please explain how and why the sample size was determined using the G-Power software in particular.
- Please address measurement uncertainty in the text, as it is not mentioned in any of the other manuscript sections.
The English could be improved to more clearly express the research.
Author Response
We sincerely thank the second reviewer for their thorough evaluation and constructive feedback on our manuscript. We have addressed each comment in detail below. For each point, we describe the changes made and indicate where in the manuscript these revisions appear.
Comment 1: The authors should describe how their work is unique and how it differs from previous studies.
- We appreciate this suggestion. We have now explicitly highlighted the novel aspects of our study and how it extends prior research. In the Introduction section, we added a sentence to clearly state that, to our knowledge, no previous human study has examined probiotic supplementation effects on endocannabinoid system markers in an physically active population. We emphasize that our trial is the first to investigate endocannabinoid (anandamide) and inflammatory responses to probiotics in female dancers, a group under high physical and psychological stress. This clarification underscores how our work differs from and builds upon prior probiotic research (page 4, line 157).
Comment 2: The statement ‘The sample size in this study was insufficient to achieve the statistical power required for detecting meaningful effects’ appears on page 11, line 405. Why wasn’t this issue taken into account before the current study was conducted?
- We acknowledge the reviewer’s concern and confirm that sample size and statistical power were carefully addressed at the study design stage. We actively pursued the recruitment goal by approaching multiple dance institutions, including academic schools, professional groups, and theaters. Despite these efforts, participant availability was severely limited by logistical challenges, such as performance schedules and sample collection constraints. Moreover, male enrollment proved insufficient. As a result, the final sample was substantially smaller (n=15). We fully recognize this limitation and have now made it explicit in the Limitations. The revised text clarifies that the study was intended to be adequately powered but ultimately proceeded as an exploratory investigation due to unresolvable recruitment barriers (page 15, line 519).
Comment 3: On page 7 of line #278, please replace Figure 1 with Figure 2.
- We have corrected this in the revised manuscript.
Comment 4: Please include the X and Y axis labels and values in Figure 2.
- We agree that Figure 2 should be fully self-explanatory. We have revised Figure 2 to incorporate clear labels and value markings on both axes. In the updated figure, the X-axis is labeled “Serum AEA (ng/mL)” and the Y-axis is labeled “Serum LPS (EU/L)”, and each axis includes appropriate numerical tick marks. This ensures that readers can readily identify the variables and units displayed on the scatter plot. We have also updated the figure caption to explicitly note the axis labels for completeness.
Comment 5: Please explain how and why the sample size was determined using the G-Power software in particular.
- G-Power is widely used in biomedical research due to its transparency, compatibility with a broad range of statistical tests, and reproducibility of results. For this study, the primary outcome was serum anandamide (AEA) concentration. Based on expected variability we assumed a medium effect size (Cohen’s d = 0.5), set α = 0.05 (two-tailed), and 80% statistical power. These parameters yielded a required sample size of 102 participants (51 per group). G-Power was selected for its strong methodological foundation, and its use in study design is supported by numerous high-quality publications. At the time of study design, GPower 1.9.7 was selected due to its broad acceptance, ease of use, and accessibility. This preceded the author’s access to institutionally licensed software such as Statistica (provided later by the Poznań University of Physical Education) which was used for further calculations.
Comment 6: Please address measurement uncertainty in the text, as it is not mentioned in any of the other manuscript sections.
- We appreciate this important observation. To address measurement uncertainty, we have revised the Methods and Discussion (Limitations subsection, page 12) to explicitly acknowledge the inherent variability and detection limitations of the ELISA-based methods used to quantify AEA, LPS, and cytokines. According to the manufacturer, the intra-assay coefficient of variation (CV) is <10%, and the inter-assay CV is <12%. We now note that minor fluctuations in biomarker concentrations may fall within the analytical variability range of the assays, potentially obscuring subtle intervention effects (page 7, line 273; page 16, line 549).
Reviewer 3 Report
Comments and Suggestions for Authors
Thank you for the opportunity to review this manuscript. The study addresses a compelling area at the intersection of gut microbiota, neuroimmunology, and sports physiology. While the topic is clinically relevant, the manuscript is significantly limited by methodological and statistical weaknesses that affect the strength of its conclusions. Below, I provide a section-by-section critique:
Title:
- Consider rephrasing to reflect the interventional and exploratory nature of the study.
Abstract:
- Please reduce emphasis on non-significant findings and clearly state that results are exploratory.
- Define the primary outcome more clearly.
Introduction:
- The rationale for choosing dancers as a study population should be strengthened—why is this population biologically relevant?
- The hypothesis is not explicitly stated. Please clarify what biological effect the intervention was expected to induce.
Methods:
- The study was designed for 102 participants but included only 15 in the final analysis. This limitation must be discussed transparently.
- There is insufficient detail on adherence, such as capsule counting or probiotic tracking.
- ELISA is not the gold standard for AEA quantification. LC-MS/MS would be more appropriate.
- The exclusion of participants post-randomization without per-protocol vs. intention-to-treat analysis introduces bias.
Results:
- ANOVA with n=5 per group is not statistically valid. Data should be presented descriptively or using non-parametric methods.
- Multiple comparisons are made, yet no correction (e.g., Bonferroni) is discussed.
- The results section should be significantly condensed, with emphasis on biologically plausible and interpretable trends.
Discussion/conclusion:
- The discussion overstates the biological relevance of non-significant findings.
- The role of the microbiota is discussed without microbiota data—this is speculative.
- Limitations regarding diet control, sample size, and ECS assay methodology must be more thoroughly discussed.
- The conclusion must be revised to reflect the exploratory, hypothesis-generating nature of the study.
Author Response
We thank the Reviewer for their thorough evaluation of our manuscript and for providing constructive and insightful feedback. We have addressed each point carefully and detail our responses below.
Comment 1: Consider rephrasing to reflect the interventional and exploratory nature of the study.
- We agree with this suggestion. The title has been revised to clearly indicate both the interventional design (randomized trial) and the exploratory scope of the study. In the new title, we explicitly include the term “exploratory randomized controlled trial” to emphasize that the findings are preliminary.
Comment 2: Please reduce emphasis on non-significant findings and clearly state that results are exploratory.
- We have condensed the abstract. The conclusion of the abstract has been rewritten to communicate that the results are preliminary and hypothesis-generating.
Comment 3: Define the primary outcome more clearly.
- We have added a sentence to the abstract to explicitly identify the primary outcome of the study.
Comment 4: The rationale for choosing dancers as a study population should be strengthened—why is this population biologically relevant?
- We appreciate the reviewer’s request for clarification. We have now expanded the Introduction to more clearly justify the biological relevance of dancers as a study population. Dancers, particularly those training and performing at a professional or semi-professional level, experience a distinct combination of physiological and psychosocial stressors. These include long, repetitive training sessions, irregular schedules, and elevated performance pressure, all of which are known to impact neuroimmune, inflammatory, and endocannabinoid signaling pathways (page 2, line 77).
Comment 5: The hypothesis is not explicitly stated. Please clarify what biological effect the intervention was expected to induce.
- We have added a clear hypothesis statement to the Introduction. Previously, our hypothesis was implied in the text; we now state it explicitly. We hypothesized that 12 weeks of L. helveticus R0052 and B. longum R0175 supplementation would beneficially modulate the endocannabinoid and inflammatory responses in the dancers. In particular, we expected the probiotic to improve gut barrier function and reduce systemic inflammation (reflected by a smaller increase in circulating LPS and pro-inflammatory cytokines), and to stabilize or reduce anandamide (AEA) levels compared to the placebo (assuming that rising AEA might indicate stress/inflammation) (page 4, line 150).
Comment 6: The study was designed for 102 participants but included only 15 in the final analysis. This limitation must be discussed transparently.
- We acknowledge the reviewer’s concern and confirm that sample size and statistical power were carefully addressed at the study design stage. We actively pursued the recruitment goal by approaching multiple dance institutions, including academic schools, professional groups, and theaters. Despite these efforts, participant availability was severely limited by logistical challenges, such as performance schedules and sample collection constraints. Moreover, male enrollment proved insufficient. As a result, the final sample was substantially smaller (n=15). We fully recognize this limitation and have now made it explicit in the Limitations. The revised text clarifies that the study was intended to be adequately powered but ultimately proceeded as an exploratory investigation due to unresolvable recruitment barriers (page 15, line 519).
Comment 7: There is insufficient detail on adherence, such as capsule counting or probiotic tracking.
- We have added details to the Methods section to describe how we monitored and ensured adherence to the supplementation regimen. In the revised text, we explain that participants were instructed to take the capsules daily and compliance was tracked through capsule counts. Participants returned their supplement containers at the last visit, and remaining capsules were counted to calculate compliance rates. We report that adherence was 100%, indicating that lack of compliance was unlikely to have influenced the results (page 5, line 229).
Comment 8: ELISA is not the gold standard for AEA quantification. LC-MS/MS would be more appropriate.
- We acknowledge the reviewer’s point regarding the assay for anandamide (AEA). In the Discussion and Limitations, we now explicitly note that we used an ELISA-based assay for AEA, and we recognize that this is less specific and sensitive than the gold-standard method (liquid chromatography–tandem mass spectrometry, LC-MS/MS) for endocannabinoid measurement. We have added a statement that this choice of assay is a methodological limitation of our study. Additionally, we mention that future studies should employ LC-MS/MS for more accurate quantification of endocannabinoids (page 14, line 441). The choice of ELISA was determined by financial considerations and the technical capabilities available at the time within the research institution. While LC-MS/MS would have offered greater analytical precision, it was not feasible within the scope and budget of this project.
Comment 9: The exclusion of participants post-randomization without per-protocol vs. intention-to-treat analysis introduces bias.
- We agree that excluding participants after randomization can introduce bias, and we have addressed this issue in Methods and Limitations. We have clarified which analyses were performed per-protocol and have acknowledged the absence of an intention-to-treat (ITT) analysis as a limitation. Previously, in the Discussion it was noted that one participant was excluded for having an outlying BMI and another for an extreme AEA value, both decisions made post-randomization. We now caution that such exclusions, while done to ensure data validity and homogeneity, may bias the results by deviating from true random assignment (page 5, line 228; page 16, line 529).
Comment 10: ANOVA with n=5 per group is not statistically valid. Data should be presented descriptively or using non-parametric methods.
- We thank the reviewer for this important observation. We agree that traditional parametric tests such as ANOVA have limited value with very small sample sizes. In the revised Results section, we no longer report ANOVA findings. Instead, we applied independent and paired t-tests where assumptions of normality were satisfied (as assessed by the Shapiro–Wilk test). For the one outcome where normality was not met, a non-parametric alternative (Mann–Whitney U test or Wilcoxon signed-rank test) was used. In addition, results are now presented descriptively, with means and ranges reported alongside p-values.
Comment 11: Multiple comparisons are made, yet no correction (e.g., Bonferroni) is discussed.
- We thank the reviewer for this observation and agree that the issue of multiple comparisons requires careful consideration. As stated in the Methods section (page 7, line 311 – “If statistically significant results …”), the possibility of applying a Bonferroni correction was anticipated in the original study design, under the assumption that statistically significant results might be observed. The study was initially powered for a much larger sample size, and our intention was to apply post hoc correction if warranted by the data. However, due to recruitment limitations, the final sample was substantially smaller than planned, and no statistically significant differences were detected across the outcomes. In this context, formal adjustment for multiple testing was not implemented, as it was deemed methodologically unnecessary and statistically impractical. The use of Bonferroni-type corrections in highly underpowered studies may obscure potentially meaningful patterns by inflating the risk of false negatives (Type II error).
Comment 12: The results section should be significantly condensed, with emphasis on biologically plausible and interpretable trends.
- In response to this suggestion, we have significantly tightened the Results section. All ANOVA results have been removed due to the small sample size and limited statistical validity of this approach in the present context. The revised section now presents data in a more concise and descriptive manner, supported by appropriate parametric or non-parametric comparisons depending on data distribution.
Comment 13: The discussion overstates the biological relevance of non-significant findings.
- We have revised the Discussion to adopt a more cautious tone regarding our findings. Any language that implied strong conclusions from non-significant results has been toned down or removed.
Comment 14: The role of the microbiota is discussed without microbiota data—this is speculative.
- We recognize that our discussion ventured into mechanisms involving the gut microbiota, even though we did not collect any microbiome data in this study. We have modified that portion of the Discussion (Limitations) to make it clear that any mention of microbiota-related effects is speculative and based on background literature, not on our own data (page 16, line 536).
Comment 15: Limitations regarding diet control, sample size, and ECS assay methodology must be more thoroughly discussed.
- We have expanded our limitations discussion to ensure all these factors are comprehensively addressed (page 15, line 515).
Comment 16: The conclusion must be revised to reflect the exploratory, hypothesis-generating nature of the study.
- We have rewritten the conclusion to clearly convey that our findings are exploratory and primarily useful for generating hypotheses. The revised conclusion now avoids any definitive statements about probiotic efficacy. Instead, we summarize that no significant effects were found and emphasize that the observed patterns are tentative. We explicitly use language such as “exploratory” and “hypothesis-generating” to characterize the study’s outcome.
Reviewer 4 Report
Comments and Suggestions for Authors
The manuscript by Wiącek et al. entitled “Assessment of Endocannabinoid and Inflammatory Markers in Dancers Following 12 Week Supplementation with Lactobacillus helveticus R0052 and Bifidobacterium longum R0175 – a randomized, placebo-controlled trial” focuses on elucidation of the protective effects of bacterial supplementation on the endocannabinoid system and its interplay with inflammation with a focus on several inflammation markers (circulating levels of cytokines). The cytokine levels are highly variable and depend on multiple parameters. Thus, it is quite challenging to exclude individuals with all conditions that might alter circulating levels of pro-inflammatory cytokines (exclusion criteria should be provided). Thus, this disadvantage should be compensated by the large sample size. Unfortunately, the sample size is too small (n=5) to draw the reliable conclusions. In addition, it is particularly emphasized that the parameters are investigated in dancers, implying that they might be influenced by physical activity. However, there is no groups with a lower physical activity to provide comparison. Furthermore, the differences were statistically insignificant, which make the conclusions quite speculative.
Abstract:
- Abbreviations should be provided in full.
Methods:
- The section should be split into subsections with individual subheading to increase the readability
- Provide commercial names of ELISA kits used
Results:
- Table 1. Use dots instead of commas
- Subheadings should reflect the key findings and be written as conclusions
- Table 2. Data should be presented as Figures (column scatter-indexed data). Moreover, the data should be presented as the Mean ± SD not as just the mean. The same should be applied to all Tables.
Author Response
We sincerely thank the fourth reviewer for their thorough evaluation and constructive feedback on our manuscript. We have addressed each comment in detail below.
Reviewer Comment 1:
“The cytokine levels are highly variable and depend on multiple parameters. Thus, it is quite challenging to exclude individuals with all conditions that might alter circulating levels of pro-inflammatory cytokines (exclusion criteria should be provided). Thus, this disadvantage should be compensated by the large sample size. Unfortunately, the sample size is too small (n=5) to draw the reliable conclusions. In addition, it is particularly emphasized that the parameters are investigated in dancers, implying that they might be influenced by physical activity. However, there is no groups with a lower physical activity to provide comparison. Furthermore, the differences were statistically insignificant, which make the conclusions quite speculative.”
We appreciate this comprehensive and important comment. Please find our point-by-point response below:
- We agree that cytokine concentrations are influenced by numerous physiological and environmental factors. To reduce potential variability, we applied strict eligibility criteria described in the Methods section (Participants subsection, page 4, line 179) and added exclusion criteria (page 4, line 183). Only individuals meeting predefined inclusion criteria were enrolled, and participants presenting with recent infections, chronic inflammatory or metabolic conditions, or taking anti-inflammatory or immunomodulatory medications were excluded. Additionally, participants were instructed to refrain from intensive physical training during the 48 hours preceding blood collection, in order to minimize acute exercise-induced fluctuations in cytokine levels.
- We fully agree that a larger sample size would be necessary to increase the statistical power of the study and allow for generalizable conclusions. We added the information in the Limitations subsection (page 15, line 519), that the small final sample (n = 15; probiotic group n = 5) is a major limitation. We explain that an a priori power analysis indicated a required sample of 102 participants to detect medium-sized effects, and that recruitment challenges (despite outreach to multiple dance schools and theaters) prevented us from reaching this target. We explicitly note in the Conclusions that the final sample size limits statistical inference and renders the study exploratory in nature.
- We acknowledge the absence of a sedentary or lower-activity control group. This limitation is now discussed in the Limitations section (page 15, line 532). The study was specifically designed to investigate physically active dancers; the inclusion of a sedentary control group was not feasible due to logistical and financial constraints, including limited recruitment resources and the targeted nature of the available study population. We now clarify that this limits the ability to isolate the effects of physical activity from the effects of the probiotic intervention.
- We agree that the lack of statistically significant findings necessitates caution in interpretation. In response, we have revised the Abstract, Limitations and Conclusions to clearly state that no significant changes were observed for any of the primary or secondary outcomes. We have also adjusted the wording throughout to reflect the hypothesis-generating nature of the findings and have removed any overstatements regarding biological relevance. All trends are now described as preliminary observations requiring confirmation in larger, adequately powered trials.
Comment 2: “Abbreviations should be provided in full.”
- This has been corrected.
Comment 3: “The section should be split into subsections with individual subheading to increase the readability.”
- The Methods section is now divided into clear, labeled subsections including: Participants, Intervention, Outcome Measures, and Statistical Analysis, which improves readability and structural clarity.
Comment 4: “Provide commercial names of ELISA kits used.”
- We thank the reviewer for the comment. The commercial names and manufacturers of all ELISA kits were provided in the original submission and remain included in the revised manuscript (Methods section, page 7, line 271). Each kit is identified by target analyte and supplier (SunRed Biotechnology, Shanghai, China), along with key technical specifications including sensitivity range and intra- and inter-assay coefficients of variation, which ensure sufficient methodological transparency.
Comment 5: “Table 1: Use dots instead of commas.”
- All decimal separators in Table 1 and subsequent tables have been converted from commas to dots, in accordance with international scientific formatting standards.
Comment 6: “Subheadings should reflect the key findings and be written as conclusions.”
- Subheadings in the Results section now reflect the main outcome categories and summarize the findings.
Comment 7: “Table 2: Data should be presented as Figures (column scatter-indexed data). Moreover, the data should be presented as the Mean ± SD not as just the mean. The same should be applied to all Tables.”
- In response to the reviewer’s comment, we have supplemented Tables 2 & 3 with a column scatter plot (now presented as Figure 3) to visualize individual-level data distribution in each group. While the scatter plot does not include overlaid means ± SD, the corresponding descriptive statistics (Mean ± SD) are retained in the table for numerical reference. Additionally, all other tables have been updated to consistently report data as Mean ± SD.
Round 2
Reviewer 2 Report
Comments and Suggestions for Authors
Accept in present form
Comments on the Quality of English LanguageThe English could be improved to more clearly express the research.
Reviewer 3 Report
Comments and Suggestions for Authors
- Despite acknowledging recruitment difficulties, the final sample size (n=15, with ~5 participants per group) is fundamentally inadequate for any inferential statistical analysis. The statistical power is effectively non-existent, rendering the results anecdotal rather than evidential. This limitation undermines the reliability and interpretability of any findings presented.
- The exclusion of participants after randomization, without the implementation of an intention-to-treat (ITT) analysis, introduces a high risk of bias. Although the authors acknowledge this limitation, their justification does not mitigate the potential distortions in the internal validity of the study.
- The absence of statistically significant results, compounded by methodological limitations, means that the manuscript does not offer novel or actionable scientific insights. The hypothetical associations discussed remain largely speculative.
- Use of ELISA for AEA quantification remains a significant limitation, as the authors admit. While financial constraints are understandable, the reliance on a suboptimal assay further diminishes confidence in the biochemical data presented.
- Lack of microbiota data renders the discussion of gut-brain axis mechanisms speculative, even though this pathway was a key component of the study’s rationale.
- In summary, despite the authors' commendable efforts to revise the manuscript and acknowledge its exploratory nature, the fundamental methodological weaknesses, primarily the inadequate sample size and post-randomization exclusions, render the study unsuitable for publication in its current form. The findings, while presented cautiously, do not advance the field in a meaningful way and are too preliminary to warrant dissemination in the peer-reviewed literature. I therefore recommend rejection of this manuscript.
Reviewer 4 Report
Comments and Suggestions for Authors
The authors have addressed the comments or provided the reasonable explanations.